# Exploring Antioxidant Properties of Standardized Extracts from Medicinal Plants Approved by the Thai FDA for Dietary Supplementation

**DOI:** 10.3390/nu17050898

**Published:** 2025-03-04

**Authors:** Surasak Limsuwan, Nurulhusna Awaeloh, Pinanong Na-Phatthalung, Thammarat Kaewmanee, Sasitorn Chusri

**Affiliations:** 1Traditional Thai Medical Research and Innovation Center, Faculty of Traditional Thai Medicine, Prince of Songkla University, Hat Yai, Songkhla 90110, Thailand; surasak.l@psu.ac.th; 2Biomedical Technology Research Group for Vulnerable Populations and School of Health Science, Mae Fah Luang University, Muang, Chiang Rai 57100, Thailand; 6451811004@lamduan.mfu.ac.th; 3Division of Hematology and Oncology, Icahn School of Medicine at Mount Sinai, New York, NY 10029, USA; pinanong.naphatthalung@mssm.edu; 4Department of Food Science and Nutrition, Faculty of Science and Technology, Prince of Songkla University, Muang, Pattani 94000, Thailand

**Keywords:** antioxidants, plant-derived functional ingredients, neuroprotective agents, Thai FDA-approved extracts, oxidative stress, dietary supplementation, functional food development

## Abstract

Background/Objectives: There is a growing interest in plant-derived antioxidants as functional food ingredients, given their potential to address oxidative stress-related diseases, notably neurodegenerative disorders. This study aims to investigate the antioxidant properties of medicinal plants that have been approved by the Thai FDA for dietary supplementation, with the goal of further utilizing them as food-functional ingredients to prevent neurodegenerative conditions. Methods: A systematic review-based methodology was employed on a list of 211 medicinal plants, and 21 medicinal plants were chosen based on their documented antioxidant activity and acetylcholinesterase (AChE) inhibitory capacity. The 21 commercially available standardized extracts were subjected to evaluation for their phenolic and flavonoid content, as well as their antioxidant activities utilizing metal-chelating activity, DPPH, ABTS free radical scavenging, ferric-reducing antioxidant power (FRAP), and superoxide anion scavenging techniques. Results: Among the 21, six extracts—*Bacopa monnieri*, *Camellia sinensis*, *Coffea arabica*, *Curcuma longa*, *Tagetes erecta*, and *Terminalia chebula*—emerged as the most promising. These extracts exhibited elevated levels of phenolic (up to 1378.19 mg gallic acid equivalents per gram) and flavonoids, with *Coffea arabica* and *Curcuma longa* showing the strongest antioxidant and free radical scavenging activities, indicating their potential for use in functional foods aimed at delaying neurodegenerative diseases. Conclusions: Due to their high levels of phenolic and flavonoid compounds, along with strong metal-chelating abilities and significant free radical scavenging activities, these standardized extracts show potential for functional food applications that may help delay the onset of neurodegenerative diseases.

## 1. Introduction

In recent years, there has been increasing interest in consuming exogenous plant-derived antioxidants and a rapidly growing demand for naturally derived functional ingredients. These antioxidants are positively associated with protective effects against several oxidative stress-mediated diseases, such as age-related conditions and chronic metabolic disorders, leading to global public health challenges [1,2]. A recent study indicates that the worldwide nutraceutical market is rising, with projections suggesting it could reach $340 billion by 2024. From 2016 to 2024, the market’s compound annual growth rate (CAGR) is estimated at 7.2% [3]. The rapid growth of the global nutraceutical market has raised several concerns regarding these products from low-to-middle-income and high-income countries [3,4].

Research has shown that a significant proportion of the global population, ranging from 10% to 80%, utilizes locally available herbs for food, health rejuvenation, and treating various illnesses [5,6,7]. It is widely recognized that medicinal plants possess highly effective antioxidant defense mechanisms, specifically through their secondary metabolites such as tannins, phenolic acids, ascorbic acid, carotenoids, flavonoids, and others [2,8]. While in vitro and in vivo studies have demonstrated the therapeutic effects and mechanisms of action of some plant-derived food ingredients [9,10], recent findings on the clinical efficacy of these nutraceuticals in terms of preventive effects remain limited. Several studies have highlighted concerns due to the varying bioavailability of different plant-derived antioxidants, emphasizing that the most abundant functional compounds must reach maximum concentrations of active metabolites in target tissues to be effective [11,12]. Additionally, nutraceuticals, often taken as supplements and available over the counter, raise significant safety concerns, and there are questions about the consistency of published results and the feasibility of scaling up for industrial production. According to previous studies, the most frequently encountered issues include contamination, adulteration (accidental or deliberate), and misleading labeling [13,14,15]. Several strategies have been implemented in response to these concerns, including the approval process for food supplements by the Ministry of Public Health or the Food and Drug Administration, including the Thai Food and Drug Administration (the Thai FDA) [16].

Thailand, known for its rich biodiversity and deep historical background in traditional medicine, has a substantial collection of medicinal plants recognized for their health-promoting properties. Additionally, the significant market in Thailand for food supplements and functional foods, coupled with its increasing aging population [17,18], requires more research to focus on locally based plants approved by the Thai FDA for dietary supplements. The Thai FDA has granted approval to 211 plant-based extracts for dietary supplementation [16]. However, their potential as functional ingredients remains underexplored. Though previous studies have shown the antioxidant and neuroprotective properties of specific medicinal plants, a comprehensive and systematic evaluation of Thai FDA-approved standardized extracts for their potential use in functional food development is lacking. Furthermore, challenges such as standardization, quality control, and consumer acceptance hinder the broader application of these extracts in commercial products.

This study aims to address the existing gap by systematically evaluating the antioxidant properties of 21 selected medicinal plants that have received approval from the Thai FDA for dietary supplements. By assessing the phenolic and flavonoid content, metal chelation activity, and free radical scavenging potential of these plants, we can identify promising candidates for incorporation into health-promoting functional foods. Furthermore, considering the significant role of oxidative stress in neurodegenerative diseases, this research establishes a foundational framework for using these extracts as neuroprotective agents. By bridging the gap between traditional knowledge and scientific validation, our findings contribute to the development of evidence-based functional food products designed to support brain health and overall well-being. The findings of this study not only advance the scientific understanding of these extracts but also have practical implications for public health and the nutraceutical market.

## 2. Materials and Methods

### 2.1. Standardized Extracts and Chemicals

The standardized extracts (*n* = 21) are described in Table 1 and are *Allium sativum*, *Aloe vera*, *Bacopa monnieri*, *Camellia sinensis*, *Capsicum annuum*, *Carthamus tinctorius*, *Centella asiatica*, *Citrus aurantium*, *Coffea arabica*, *Curcuma longa*, *Daucus carota*, *Ganoderma lucidum*, *Garcinia mangostana*, *Gynostemma pentaphyllum*, *Kaempferia parviflora*, *Matricaria chamomilla*, *Moringa oleifera*, *Piper nigrum*, *Tagetes erecta*, *Terminalia chebula*, and *Zingiber officinale*. These standardized extracts were obtained from the AP Operations Co., Ltd., Chonburi, Thailand. Unless stated elsewhere, each extract was freshly dissolved in distilled water or 95% ethanol to test their biological activities.

For all assays, analytical grade reagents including gallic acid, catechin, 6-hydroxy-2,5,7,8-tetramethylchroman-2-carboxylic acid (Trolox), potassium ferricyanide, ferric chloride, aluminum chloride, nitro blue tetrazolium (NBT), 2,2′-azino-bis-(3-ethylbenzothiazoline-6-sulphonic acid) (ABTS), ferric chloride (FeCl_3_), potassium persulphate, 2,2′-diphenyl-1-picrylhydrazyl (DPPH), and Folin–Ciocalteu were obtained from Merck (Merck KGaA, Darmstadt, Germany) or Sigma (Sigma-Aldrich Chemie, MO, USA). 

### 2.2. A Systematic Review-Based Approach for Identifying Potent, Reliable, and Highly Feasible Medicinal Plant Extracts, as Described in the Thai FDA Approval List for Dietary Supplementation

The medicinal plants examined in this study (*n* = 211) were thoroughly documented in the approval list by the Thai FDA for their use as dietary supplements [16]. To conduct a comprehensive exploration of locally sourced functional ingredients with neuroprotective properties, specific inclusion criteria were followed in selecting candidate medicinal plants (Figure 1). The first inclusion criterion was that only domestically cultivated plants were considered to ensure the local availability of herbal products. Secondly, the plants were selected based on their reported effectiveness in scavenging DPPH and ABTS radicals, which are well-established methodologies. The rationale for selection included the widely recognized fact that oxidative damage from excess free radicals significantly contributes to neurodegenerative diseases. Third, available information on their acetylcholinesterase (AChE) inhibitory capacity, a recognized screening approach for plants with neuroprotective potential, was included in the selection criteria. Finally, plants with commercially available standardized extracts were considered to verify their feasibility for use in the functional food industry. Detailed information on the 211 medicinal plants, including their scientific names, utilized plant parts, antioxidant properties, and AChE inhibitory effects, is provided in Appendix A, respectively.

### 2.3. Measurement of Total Phenolics and Flavonoids Content of Promising Standardized Extracts

The Folin–Ciocalteu (FC) method was employed with slight modification. During the experiments, the reagents and sample solutions were freshly prepared, including the FC reagent, which was diluted to 1:10 with distilled water, the solution of sodium carbonate (Ajax Finechem, Auckland, New Zealand; 20% *w*/*v*) in distilled water, and the solutions of each standardized extract and a standard compound, gallic acid (Sigma-Aldrich Co., St. Lovis, MO, USA) (1000–0.5 µg/mL) were two-fold serially dissolved in 95% (*v*/*v*) ethanol. To quantify the TPC of the extracts, 120 µL of each extract was mixed with 1000 µL of the FC reagent for 5 min, and then an aliquot of 1000 µL of sodium carbonate solution was added to this mixture. The solution was thoroughly mixed and was incubated in the dark at 25 °C for 1.5 h. The absorbance was measured at 725 nm (Sunrise™ Microplate reader, Tecan Group Ltd., Männedorf, Switzerland). The TPC was reported as the milligrams of gallic acid (Sigma-Aldrich Chemie, Darmstadt, Germany) per gram of extracts [19].

The aluminum chloride colorimetric method with slight modification was applied to quantify the total flavonoid content (TFC) of each extract [19]. In the aluminum chloride colorimetric assay, the fundamental concept involves the formation of acid-stable complexes between aluminum chloride and the C-4 keto group as well as either the C-3 or C-5 hydroxyl group of flavones and flavonols. Furthermore, it also facilitates the creation of acid-labile complexes with the ortho-dihydroxyl groups located in the A- or B-ring of flavonoids. In brief, a 50 µL aliquot of each extract was combined with 300 µL of 5% (*w*/*v*) sodium nitrite (Ajax Finechem, Auckland, New Zealand) and 300 µL of 10% (*w*/*v*) aluminum trichloride (Ajax Finechem, Auckland, New Zealand), then followed by the addition of 4 mL of distilled water. This mixture was kept at 25 °C for 6 min. Subsequently, 2 mL of 1 M sodium hydroxide solution was introduced to cease the reaction. The resultant volume was adjusted to 10 mL with distilled water, and the absorbance at 510 nm was measured after 10 min of incubation at the mentioned condition. The catechin (Sigma-Aldrich Co., St. Lovis, MO, USA) concentration found in each extract was calculated from the regression equation using their absorbance. The results were converted to the TFC as mg of catechin equivalent per gram of extract.

### 2.4. Metal-Chelating Activity

The metal-chelating technique was utilized to evaluate the capacity of each extract to chelate ferrous ions. A two-fold serial dilution of the extract (0.03–62.50 mg/mL; 250 µL) was freshly prepared and combined with 25 µL of iron (II) chloride (2 mM) and 80 µL of distilled water. Subsequently, 50 µL of ferrozine (5 mM) was added to initiate the reaction, and the resulting solution was promptly incubated at 25 °C for 10 min. The absorbance of the resulting solution was measured at 562 nm. Ethylenediaminetetraacetic acid (EDTA) (HiMedia Laboratories Pvt. Ltd., Nashik, India) served as a standard metal chelator. The chelating ability percentage was calculated using the formula: MCA (%) = [(OD control − OD sample)/OD control] × 100. This activity was expressed as the 50% inhibition concentration of the ferrous ion-ferrozine complex (IC_50_ mg/mL) for both the standard EDTA and the extracts [20,21].

### 2.5. DPPH and ABTS Free Radical Scavenging Activity

The DPPH and ABTS assays are among the most widely utilized techniques for determining the antioxidant activity of plant-derived compounds. The principle of the DPPH assay relies on reducing the purple DPPH· to 1,1-diphenyl-2-picryl hydrazine, which is dissolved in organic media; hence, this technique is suitable for antioxidants dissolved in hydrophobic systems. On the other hand, the ABTS assay is conducted based on the reduction of a blue/green ABTS^•+^, a reaction applicable to both hydrophilic and lipophilic antioxidant systems.

The DPPH assay, conducted based on a previously published procedure with slight modifications, aimed to evaluate the scavenging abilities of the extracts against the stable radical 2,2-diphenyl-1-picrylhydrazyl. Extract solutions were prepared as two-fold dilutions of each sample ranging from 2500 to 1.22 µg/mL. A 20 µL aliquot of each concentration was transferred to a pre-filled 96-well plate containing 180 µL of DPPH solution at a concentration of 80 µM and thoroughly mixed for 5 min. The decolorization of the DPPH solution was measured by recording the absorbance at 492 nm after 30 min of incubation at 25 °C in the dark.

ABTS^•+^ radicals were generated following an established protocol by combining equal volumes of 2,2′-azino-bis-3-ethylbenzothiazoline-6-sulfonic acid (prepared at a concentration of 7 mM) and potassium persulfate (at a concentration of 2.45 mM) as the substrate and oxidant, respectively. The freshly prepared ABTS^+^ solution was obtained after 12 h of incubation at 25 °C in the dark and diluted with ethanol to achieve an absorbance of 0.70 ± 0.05 at 734 nm. Subsequently, a 200 µL aliquot of this solution was transferred to a 96-well microplate containing 20 µL of the serially diluted extracts (ranging from 2500 to 1.22 µg/mL). The plate was immediately incubated at 25 °C for 6 min, and the absorbance at 734 nm was measured after 6 min of incubation at the same temperature.

The free radical scavenging capacities of both the positive control Trolox (Sigma-Aldrich Co., St. Lovis, MO, USA) and the extracts are expressed as the percentage of their inhibitory effect against the radical, along with the 50% inhibitory concentration (IC_50_; mg/mL) of the radicals [20,21].

### 2.6. Ferric-Reducing Antioxidant Power

Following a previously established protocol, the total antioxidant capacity of the extracts was evaluated through the ferric-reducing ability of plasma (FRAP) assay. The antioxidant efficacy of the extracts is demonstrated by their capacity to reduce the ferric ion within the ferric-tripyridyl triazine (TPTZ) complex to ferrous-TPTZ. In summary, a freshly prepared FRAP reagent (1.35 mL) was combined with a 150 µL aliquot of the extract solution, prepared as a two-fold serial dilution. This mixture was maintained at 37 °C for 2 h, and the increment in absorption of the solution at 593 nm was recorded. Subsequently, the absorbance values were plotted against the concentrations of ferrous ions in the ethanol solution. The ability of the extract to reduce the ferric ion is expressed as mM of ferrous per mg of extract (µM Fe_2_SO_4_/mg extract), utilizing a Fe_2_SO_4_ (Elago Enterprises Pty Ltd., New South Wales, Australia) working solution as a standard curve [20,21].

### 2.7. Superoxide Anion Scavenging Method

To determine the superoxide anion scavenging activity of the extracts, the superoxide ion was photochemically generated through a combined system of riboflavin, methionine, and illumination. The activity was assessed by measuring the inhibitory effect on forming a purple-colored formazan (NBT^2+^). To initiate the reaction, a working solution (400 µL) containing riboflavin (30 µg/mL), methionine (30 µg/mL), and EDTA (20 µg/mL) was prepared and mixed with 100 µL of NBT (400 µg/mL). Antioxidant agents, either a freshly prepared two-fold serial dilution extract or the reference substance, catechin (Sigma-Aldrich Co., St. Lovis, MO, USA), prepared in phosphate buffer (0.05 M, pH 7.4), were added to this working solution. Subsequently, light induction was achieved using fluorescent lamps (20 W) at 25 °C for 25 min, and the quantity of produced formazan was measured at 560 nm. The ability of the extracts to scavenge superoxide ions was determined as the concentration of extract required to inhibit 50% of the superoxide radical (IC_50_; mg/mL), calculated from dose-inhibition curves [19].

### 2.8. Statistical Analysis

The data are presented as the mean ± standard deviation from three experiments. Statistical analyses were conducted using SPSS™ software v.19.0 (IBM Corp. Released 2010, IBM SPSS Statistics for Windows, and Version 19.0. Armonk, NY, USA: IBMCorp). Data were assessed for statistical significance utilizing Dunnett’s test. Pearson’s correlation was performed using Microsoft Excel 2013. Heatmap analysis was performed to exhibit the antioxidant capacities of the extract samples using the R statistical system (http://www.R-project.org/, version 4.1.0, accessed on 19 September 2023).

## 3. Results

Among the 211 medicinal plant extracts listed in the Thai FDA approval list for dietary supplementation, screening was performed based on the viability of their published data, reflecting their antioxidant activity, neuroprotective potency, reliability, and feasibility. Approximately 70% of these plants are domestically cultivated (Figure 2). Of the 211 medicinal plant extracts listed in the Thai FDA approval list for dietary supplementation, the majority belong to Asteraceae (*n* = 12) and Rosaceae (*n* = 12), followed by Lamiaceae (Labiatae) (*n* = 10) and Fabaceae (*n* = 10), respectively. Approximately 70% of these plants are domestically cultivated (Figure 2). Although 116 plants (approximately 55% of all plants listed) have been reported for their antioxidant activity against DPPH and ABTS radicals, the data obtained from 54 plants show no correlation between the results obtained from the DPPH and ABTS assays, and 34 plants have not been tested for their AChE inhibitory effects. Most of the tested plants with reported activity against DPPH and ABTS radicals were classified as either strong or very strong antioxidants. In contrast, only a few plants exhibit strong inhibitory effects against AChE (Figure 2). Despite 116 plants showing antioxidant activity, only 21 extracts were commercially standardized with certificates verifying active constituents.

The total phenolic content (TPC) of the 21 extracts ranged from 165.02 to 1378.19 mg gallic acid equivalents per gram, with *Coffea arabica* having the highest value, followed by *Curcuma longa*. Flavonoid content (TFC) ranged from 11.95 to 521.41 mg catechin equivalents per gram. *Coffea arabica* again led in TFC, followed by *Curcuma longa* and *Bacopa monnieri* (Table 2). Among the tested extracts, the standardized extract obtained from the seeds of *Coffea arabica* possesses the highest phenolic content, which was found to be two times higher than that of the subsequent extract, *Curcuma longa*. Other noteworthy extracts that warrant attention include *Camellia sinensis*, *Bacopa monnieri*, *Allium sativum*, and *Garcinia mangostana*, which were also found to contain high levels of phenolic compounds ranging from 241.74 to 387.43 mg gallic equivalence per gram of extract. Similar to the results obtained from the TPC assay, the extract from *Coffea arabica* exhibits the highest content of flavonoids, followed by *Curcuma longa*, *Bacopa monnieri*, and *Allium sativum*. The subsequent extracts, namely *Aloe vera*, *Capsicum annuum*, *Daucus carota*, *Ganoderma lucidum*, *Moringa oleifera*, *Terminalia chebula*, and *Zingiber officinale*, contain moderately high levels of flavonoids ranging between 30–50 mg catechin equivalence per gram of extract.

Antioxidant activity is a multifaceted process typically involving several mechanisms and influenced by numerous factors. Thus, it is impractical to capture it thoroughly with a single method. Hence, multiple assessments were used to test an overview of the antioxidant activities of these extracts. At the 1 mg/mL tested concentration, 13 of 21 standardized extracts exhibited more than 90% metal-chelating inhibitory activity, as presented in Table 3. *Aloe vera*, *Capsicum annuum*, *Ganoderma lucidum*, *Matricaria chamomilla*, *Moringa oleifera*, and *Tagetes erecta* were identified as plants with the most potent metal-chelating properties. The FRAP assay measures the reducing potential of an antioxidant, wherein free radical chain breaking occurs through donating a hydrogen atom. As summarized in Table 3, the FRAP values ranged from 0.91 to 138.88 µM FeSO_4_/mg extract. The highest value was observed for *Coffea arabica* (138.88 ± 7.13 µM FeSO_4_/mg extract) and the lowest for *Capsicum annuum* (0.91 ± 0.02 µM FeSO_4_/mg extract). The rank order of reducing power was consistent with that of the TPC and TFC, namely, in decreasing order: *Coffea arabica*, *Curcuma longa*, *Bacopa monnieri*, and *Camellia sinensis*. Additionally, some extracts exhibited promising reducing potential, including *Allium sativum*, *Aloe vera*, *Carthamus tinctorius*, *Centella asiatica*, *Citrus aurantium*, *Daucus carota*, *Ganoderma lucidum*, *Garcinia mangostana*, *Gynostemma pentaphyllum*, and *Terminalia chebula*.

As depicted in Table 4, the percentage of free radical scavenging of each extract was tested at the concentration of 1 mg/mL against DPPH, ABTS, and superoxide radicals. All extracts reduced the stable, purple-colored radical DPPH into yellow-colored DPPH-H at different levels, ranging from 3.35% to 90–100%. Among the tested samples, the standardized extracts obtained from *Camellia sinensis*, *Coffea arabica*, *Curcuma longa*, and *Tagetes erecta* were the most potent DPPH radical scavengers. Similarly, in the ABTS assay, extracts from *Camellia sinensis*, *Coffea arabica*, *Curcuma longa*, and *Terminalia chebula* were the most effective components for scavenging free radicals. Regarding superoxide-free radical scavenging activity testing by the NBT assay, *Camellia sinensis*, *Coffea arabica*, *Curcuma longa*, and *Terminalia chebula* exhibited potent activity, with the percentage of free radical inhibition ranging from 80 to 100%.

In terms of antioxidant activity and the contents of antioxidant-related constituents illustrated in Figure 3, six standardized extracts from *Bacopa monnieri*, *Camellia sinensis*, *Coffea arabica*, *Curcuma longa*, *Tagetes erecta*, and *Terminalia chebula* demonstrated potent antioxidant capabilities. Consequently, these extracts underwent further examination to determine their IC_50_ against DPPH, ABTS, and superoxide radicals (Table 5). There is no significant difference in their metal-chelating potency, with the IC_50_ ranging from 0.06 mg/mL in *Curcuma longa* to 1.23 mg/mL observed in *Camellia sinensis*. *Coffea arabica* and *Curcuma longa* extracts exhibited remarkable free radical scavenging activity against DPPH and ABTS, with IC_50_ values ranging from 0.17 to 0.42 mg/mL and 0.12 to 0.18 mg/mL, respectively. On the other hand, extracts from *Camellia sinensis*, *Tagetes erecta*, and *Terminalia chebula* demonstrated a strong superoxide anion scavenging effect, with IC_50_ values ranging from 0.06 to 0.13 mg/mL.

## 4. Discussion

In recent years, growing awareness of the role of oxidative stress in brain health has highlighted the potential of antioxidants, particularly dietary antioxidants, as prevention strategies for neurodegenerative diseases. This study systematically evaluated standardized extracts from Thai FDA-approved medicinal plants, focusing on their antioxidant and neuroprotective potential. The inclusion criteria were: (I) plants cultivated domestically were included to ensure the local availability of herbal products; (II) their efficacy in scavenging free radicals was assessed, considering their role in the onset of neurodegenerative diseases and evaluated using widely accepted methods such as DPPH and ABTS assays; (III) the inhibitory capacity against AChE was examined, which serves as a recognized screening method for plants with neuroprotective potential; and (IV) commercially available standardized extracts were included to evaluate the practicality of the identified medicinal plants for use in the functional food industry. However, it should be noted that 21 potential medicinal plants were selected based on their availability as standardized extracts in the market. Some plants, such as *Alpinia galanga*, *Beta vulgaris*, *Cynara scolymus*, *Fragaria ananassa*, *Garcinia atroviridis*, *Illicium verum*, and *Origanum vulgare*—reported to have a notable AChE inhibitory effect (detailed in Appendix A)—were excluded from this study due to the lack of commercially available extracts. Further studies are urgently needed on the large-scale production of standardized extracts to enhance and support the utilization of medicinal plants in the food industry.

Six promising extracts—*Bacopa monnieri*, *Camellia sinensis*, *Coffea arabica*, *Curcuma longa*, *Tagetes erecta*, and *Terminalia chebula*—were identified based on their high phenolic and flavonoid content, potent metal-chelating activity, and free radical scavenging capabilities. Although their antioxidant capacities are lower than those of commercially available antioxidants such as butylated hydroxytoluene [22,23], ascorbic acid [23,24], α-tocopherol [24], and quercetin [24], these plants have been reported to exhibit AChE inhibitory effects, which are considered fundamental for the onset of neurodegenerative diseases. Among the tested plants, *Coffea arabica* stood out for its exceptionally high phenolic and flavonoid content, contributing to its superior antioxidant activity. *Curcuma longa* exhibited strong metal-chelating properties and potent free radical scavenging, attributed to its curcuminoid content. *Bacopa monnieri* demonstrated significant neuroprotective effects through its free radical scavenging activity, which aligns with its traditional use in improving cognitive function [25,26]. *Camellia sinensis*, *Tagetes erecta*, and *Terminalia chebula* showed complementary antioxidant mechanisms, further supporting their use as functional food ingredients. Even though both *Bacopa monnieri* and *Curcuma longa* are widely used as supplements, their utilization in functional food industries remains limited due to certain drawbacks, as outlined below.

*Bacopa monnieri*, commonly known as Brahmi, is a well-recognized Ayurvedic medicine esteemed for its memory and learning enhancement properties [25,26]. Numerous robust cell-based and in vivo experiments have demonstrated that *Bacopa monnieri*, along with its principal bioactive constituents—such as bacoside A, bacoside B, bacogenin A, bacopaside I, bacopasaponin C, cucurbitacin B, bacosine, and stigmasterol—exhibits neuroprotective effects and therapeutic potential against neurodegenerative diseases [27,28]. In line with the findings of this study, *Bacopa monnieri* is known as a potent antioxidant, and its promising free radical scavenging ability contributes to its neuroprotective effects. This mechanism of action has been confirmed in animal models, demonstrating the reversal of amnesic effects induced by neurotoxins, including methamphetamine, 1-methyl-4-phenylpyridinium, scopolamine, and phenytoin [29,30,31]. Although earlier research suggested that this plant could be valuable for alleviating neurodegeneration induced by oxidative stress, systematic reviews and meta-analyses have failed to confirm its cognitive-enhancing effects and therapeutic properties in treating dementia due to Alzheimer’s disease [25,32]. Interestingly, some results from a meta-analysis suggest that consumption of *Bacopa monnieri* for 12 consecutive weeks in healthy volunteers or volunteers with memory impairment complaints leads to improved cognition, particularly as evidenced by a shortened Trail B test and decreased choice reaction time [26]. In summary, *Bacopa monnieri* should be highlighted as a potential herbal product in the functional food industry. However, its bitter taste, stability of active ingredients, and reliability as a neuroprotective food require intensive further study.

Another promising medicinal plant identified in this study is *Curcuma longa*. The primary active constituents of this plant are curcuminoids, comprising 60–70% curcumin, 20–27% demethoxycurcumin, and 10–15% bisdemethoxycurcumin, which are present in approximately 1–6% of its rhizome dry-weight [33]. The neuroprotective properties of curcumin are attributed to its abilities in metal chelation, antioxidant activity, anti-inflammatory effects, anti-apoptotic actions, reduction of amyloid plaque formation, and enhancement of neurogenesis [34,35,36,37]. These effects have been demonstrated in several animal models, including Parkinson’s disease (PD) [38], amyotrophic lateral sclerosis (ALS) [39], and Alzheimer’s disease (AD) [40]. Even though recent systematic reviews of clinical studies have highlighted that curcumin supplementation increased serum brain-derived neurotrophic factor (BDNF) levels, curcumin showed no beneficial effects in patients with AD [41]. Furthermore, some adverse events, primarily related to gastrointestinal disorders, have been reported in certain clinical studies, raising awareness about the potential risks of regular curcumin consumption as a functional food [41]. Several aspects, particularly the effectiveness and safety of this compound as a neuroprotective food, need to be clarified in further clinical studies.

## 5. Conclusions

This study conducted a systematic evaluation of the antioxidant properties of Thai FDA-approved medicinal plant extracts, identifying *Bacopa monnieri*, *Camellia sinensis, Coffea arabica*, *Curcuma longa*, *Tagetes erecta*, and *Terminalia chebula* as the most promising candidates. These extracts displayed high levels of phenolic and flavonoid compounds, strong metal-chelating ability, and significant free radical scavenging activity, indicating their potential in functional food applications. The results provide valuable insights into the antioxidant capabilities of these plant-derived substances, laying a scientific groundwork for their use in nutraceutical and functional food products. Future research should prioritize bioavailability, long-term safety, and formulation strategies to enhance their practical application in consumer health items.

## Figures and Tables

**Figure 1 nutrients-17-00898-f001:**
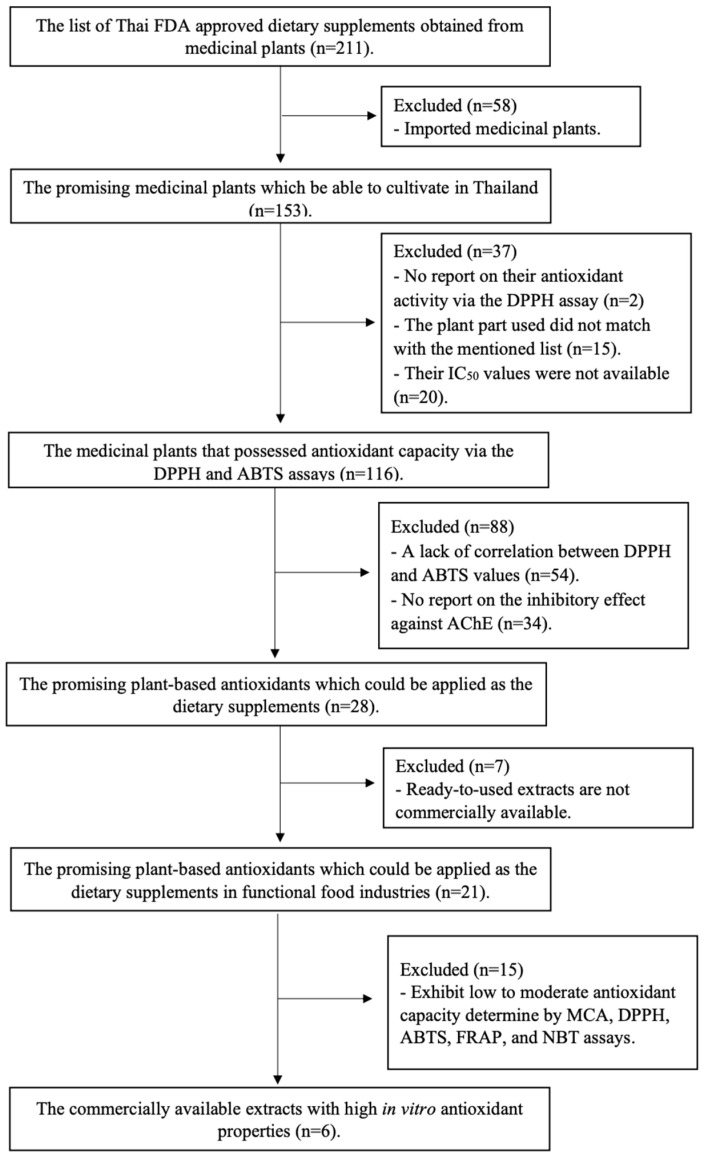
The flow diagram for the systematic review-based approach to screening for potent neuroprotective agents in Thai FDA-approved dietary supplements derived from medicinal plants.

**Figure 2 nutrients-17-00898-f002:**
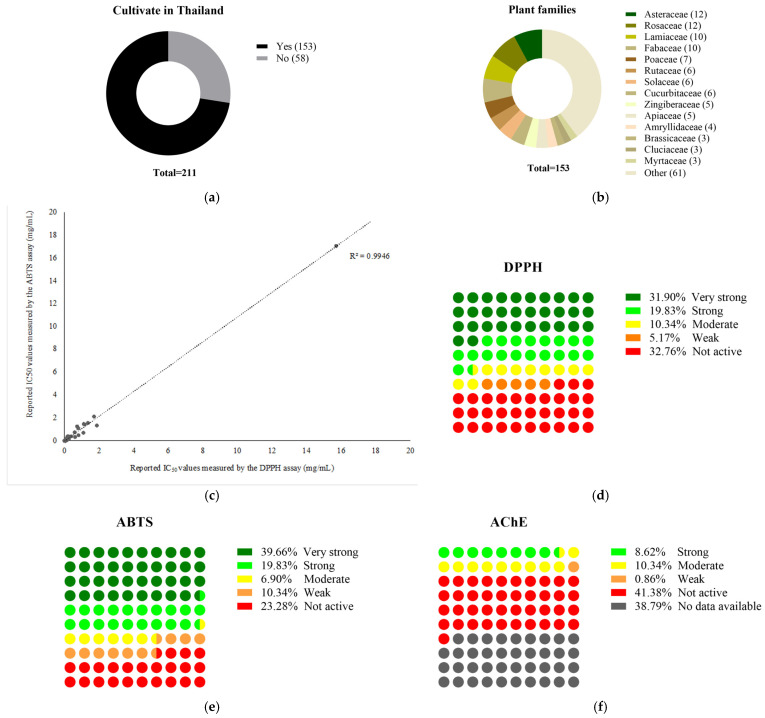
Based on the obtained data, medicinal plants described in the list of Thai FDA-approved dietary supplements can either be domestically cultivated herbs or imported plant parts (**a**), belonging to various families, with Asteraceae and Rosaceae being the majority (**b**). Intensive studies focus on their free radical scavenging effects on DPPH and ABTS radicals (**c**–**e**). In vitro studies on their neuroprotective biological activities have widely investigated their inhibitory effects against acetylcholinesterase (AChE), which is a cholinergic enzyme involved in the hydrolysis of the neurotransmitter acetylcholine and serves as a strategy for the treatment of Alzheimer’s disease (**f**).

**Figure 3 nutrients-17-00898-f003:**
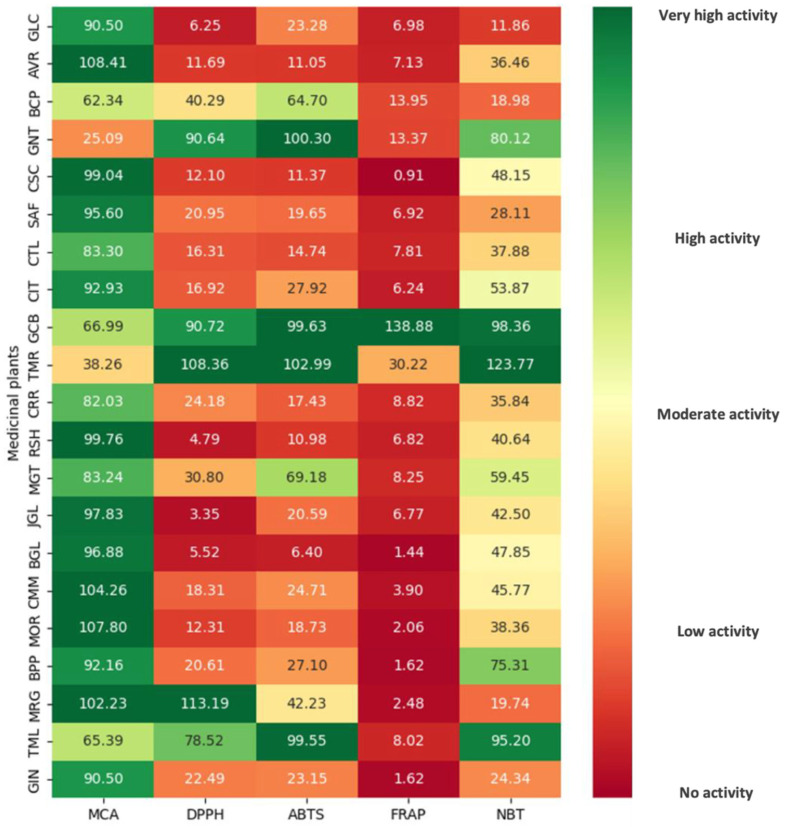
A heatmap analysis reveals varying levels of antioxidant activities among selected standardized extracts from a list of Thai FDA-approved dietary supplements derived from medicinal plants. Green indicates the highest level, yellow represents the average level, and red signifies the lowest level. The values in the heatmap reflect the percentage of free radical scavenging for each extract, evaluated through DPPH, ABTS, and NBT assays, along with their antioxidant capacity, as shown by the FRAP values. It also includes the percentage of metal-chelating inhibition evaluated using the MCA assay. The tested extracts include *Allium sativum* (GLC), *Aloe vera* (AVR), *Bacopa monnieri* (BCP), *Camellia sinensis* (GNT), *Capsicum annuum* (CSC), *Carthamus tinctorius* (SAF), *Centella asiatica* (CTL), *Citrus aurantium* (CIT), *Coffea arabica* (GCB), *Curcuma longa* (TMR), *Daucus carota* (CRR), *Ganoderma lucidum* (RSH), *Garcinia mangostana* (MGT), *Gynostemma pentaphyllum* (JGL), *Kaempferia parviflora* (BGL), *Matricaria chamomilla* (CMM), *Moringa oleifera* (MOR), *Piper nigrum* (BPP), *Tagetes erecta* (MRG), *Terminalia chebula* (TML), and *Zingiber Officinale* (GIN).

**Table 1 nutrients-17-00898-t001:** Commercially details some promising medicinal plants chosen based on their reported antioxidant activity and acetylcholinesterase inhibitory effects.

No.	Botanical Name	Bioactive Markers/Analysis Items (%; *w*/*w*)		
		Compounds	Required Amount	Reported Amount
1.	*Allium sativum*	Allicin	>1.00	1.20
2.	*Aloe vera*	Aloin content	NA	≤0.1 ppm
3.	*Bacopa monnieri*	Bacosides	≥20.00	21.24
4.	*Camellia sinensis*	Polyphenols	>10.00	10.40
5.	*Capsicum annuum*	Capsaicin	≥10.00	12.40
6.	*Carthamus tinctorius*	Flavonoids	>300.00	375.10
7.	*Centella asiatica*	Asiaticoside	>2.50	2.82
		Triterpene	>1.00	1.74
8.	*Citrus aurantium*	Hesperidin	>99.00	99.91
9.	*Coffea arabica*	Chlorogenic acid	≥50.00	50.25
10.	*Curcuma longa*	Curcumin	>5.00	7.40
11.	*Daucus carota*	Beta-carotene	≥10.00	13.67
12.	*Ganoderma lucidum*	Polysaccharides	>50.00	72.80
		Triterpenoids	>1.00	1.50
13.	*Garcinia mangostana*	Xanthone	>1.00	1.30
14.	*Gynostemma pentaphyllum* Saponins	≥1000.00	1392.00	
15.	*Kaempferia parviflora*	Flavonoids	>1.50	1.51
16.	*Matricaria chamomilla*	Apigenin	>1.00	1.26
17.	*Moringa oleifera*	Flavonoids	>500.00	567.10
18.	*Piper nigrum*	Piperine	≥10.00	10.20
19.	*Tagetes erecta*	Lutein	≥10.00	10.26
20.	*Terminalia chebula*	Gallic acid	≥500.00	603.90
21.	*Zingiber officinale*	Gingerols	≥5.00	5.21

NA: Not available.

**Table 2 nutrients-17-00898-t002:** Total phenolic and flavonoid content of selected standardized extracts from the list of Thai FDA-approved dietary supplements derived from medicinal plants.

Medicinal Plants	Part Used	Total Contents of (mg Equivalence/g of Extract)	
		Phenolic	Flavonoid
*Allium sativum* (GCL)	Rhizome	242.52 ± 17.14 ^de^	70.22 ± 11.79 ^d^
*Aloe vera* (AVR)	Gel	211.52 ± 15.78 ^defgh^	35.41 ± 2.85 ^gh^
*Bacopa monnieri* (BCP)	Whole tree	258.02 ± 12.80 ^d^	80.35 ± 3.95 ^c^
*Camellia sinensis* (GNT)	Fruit	387.43 ± 31.26 ^c^	15.79 ± 1.10 ^j^
*Capsicum annuum* (CSC)	Leaf	180.14 ± 10.55 ^fgh^	36.36 ± 2.51 ^fg^
*Carthamus tinctorius* (SAF)	Flower	222.37 ± 11.09 ^defg^	28.45 ± 6.10 ^ghi^
*Centella asiatica* (CTL)	Leaves	210.75 ± 21.78 ^defgh^	1.65 ± 0.16 ^k^
*Citrus aurantium* (CIT)	Fruit	165.02 ± 8.25 ^h^	15.16 ± 1.45 ^j^
*Coffea arabica* (GCB)	Seeds	1378.19 ± 85.34 ^a^	521.47 ± 10.16 ^a^
*Curcuma longa* (TMR)	Rhizome	514.13 ± 50.55 ^b^	185.22 ± 8.87 ^b^
*Daucus carota* (CRR)	Root	215.78 ± 17.45 ^defgh^	34.15 ± 3.95 ^gh^
*Ganoderma lucidum* (RSH)	Fruit	205.32 ± 16.94 ^defgh^	33.51 ± 0.95 ^gh^
*Garcinia mangostana* (MGT)	Peel	241.74 ± 16.82 ^de^	11.95 ± 1.11 ^j^
*Gynostemma pentaphyllum* (JGL)	Leaves	203.38 ± 21.19 ^defgh^	28.45 ± 2.19 ^ghi^
*Kaempferia parviflora* (BGL)	Rhizome	200.28 ± 10.65 ^efgh^	27.82 ± 1.64 ^hi^
*Matricaria chamomilla* (CMM)	Flower	169.67 ± 13.87 ^gh^	26.55 ± 1.45 ^ghij^
*Moringa oleifera* (MOR)	Leaf	204.16 ± 14.53 ^defgh^	32.56 ± 1.64 ^gh^
*Piper nigrum* (BPP)	Seed	230.51 ± 5.07 ^def^	20.22 ± 1.90 ^ij^
*Tagetes erecta* (MRG)	Flower	173.94 ± 8.16 ^gh^	13.26 ± 0.55 ^j^
*Terminalia chebula* (TML)	Fruit	215.01 ± 5.85 ^defgh^	43.96 ± 0.95 ^f^
*Zingiber officinale* (GIN)	Rhizome	218.49 ± 17.14 ^defgh^	51.87 ± 3.84 ^e^

^a–k^: Values in the same row with different superscripts are significantly different (*p* < 0.05).

**Table 3 nutrients-17-00898-t003:** Metal-chelating activity (MCA) and ferric-reducing antioxidant power (FRAP) of selected standardized extracts from the list of Thai FDA-approved dietary supplements derived from medicinal plants.

Medicinal Plants	Metal-Chelating Activity	Ferric-Reducing Antioxidant Power
	(%Inhibition ± SD at conc. 1 mg/mL)	(µM Fe_2_SO_4_/mg)
*Allium sativum* (GCL)	90.50 ± 6.98 ^efg^	6.98 ± 0.19 ^d^
*Aloe vera* (AVR)	108.41 ± 4.41 ^a^	7.13 ± 0.70 ^d^
*Bacopa monnieri (BCP)*	62.34 ± 4.90 ^h^	13.95 ± 0.35 ^c^
*Camellia sinensis* (GNT)	25.09 ± 1.28 ^j^	13.37 ± 0.22 ^c^
*Capsicum annuum* (CSC)	99.04 ± 0.96 ^bcde^	0.91 ± 0.02 ^f^
*Carthamus tinctorius* (SAF)	95.60 ± 4.60 ^cde^	6.92 ± 0.15 ^de^
*Centella asiatica* (CTL)	83.30 ± 5.71 ^fg^	7.81 ± 0.15 ^d^
*Citrus aurantium* (CIT)	92.93 ± 4.83 ^de^	6.24 ± 0.65 ^de^
*Coffea arabica* (GCB)	66.99 ± 6.23 ^h^	138.88 ± 7.13 ^a^
*Curcuma longa* (TMR)	38.26 ± 2.49 ^i^	30.22 ± 2.27 ^b^
*Daucus carota* (CRR)	82.03 ± 1.72 ^g^	8.82 ± 0.17 ^d^
*Ganoderma lucidum* (RSH)	99.76 ± 1.04 ^abcde^	6.82 ± 0.06 ^de^
*Garcinia mangostana* (MGT)	83.24 ± 7.66 ^fg^	8.25 ± 0.58 ^d^
*Gynostemma pentaphyllum* (JGL)	97.83 ± 1.63 ^cde^	6.77 ± 0.02 ^de^
*Kaempferia parviflora* (BGL)	96.88 ± 0.22 ^cde^	1.44 ± 0.11 ^f^
*Matricaria chamomilla* (CMM)	104.26 ± 9.12 ^abc^	3.90 ± 0.19 ^ef^
*Moringa oleifera* (MOR)	107.80 ± 9.80 ^ab^	2.06 ± 0.08 ^f^
*Piper nigrum* (BPP)	92.16 ± 3.01 ^ef^	1.62 ± 0.05 ^f^
*Tagetes erecta* (MRG)	102.23 ± 4.98 ^abcd^	2.48 ± 0.11 ^f^
*Terminalia chebula* (TML)	65.39 ± 4.34 ^h^	8.02 ± 0.17 ^d^
*Zingiber officinale* (GIN)	90.50 ± 2.98 ^efg^	1.62 ± 0.10 ^f^

^a–j^: Values in the same row with different superscripts are significantly different (*p* < 0.05).

**Table 4 nutrients-17-00898-t004:** Free radical scavenging properties of selected standardized extracts from the list of Thai FDA-approved dietary supplements derived from medicinal plants.

Medicinal Plants	Free Radical Scavenging Activities Against (%Inhibition ± SD at conc. 1 mg/mL)		
	DPPH Radicals	ABTS Radicals	Superoxide Radicals
*Allium sativum*	6.25 ± 0.58 ^l^	23.28 ± 0.67 ^h^	11.86 ± 0.68 ^n^
*Aloe vera*	11.69 ± 0.44 ^k^	11.05 ± 0.52 ^l^	32.46 ± 1.40 ^jk^
*Bacopa monnieri*	40.29 ± 2.05 ^e^	64.70 ± 0.93 ^d^	18.98 ± 1.34 ^m^
*Camellia sinensis*	90.64 ± 0.04 ^c^	100.30 ± 0.07 ^b^	80.12 ± 4.79 ^c^
*Capsicum annuum*	12.10 ± 0.71 ^k^	11.37 ± 1.06 ^l^	48.15 ± 2.69 ^g^
*Carthamus tinctorius*	19.65 ± 0.48 ^ij^	6.92 ± 0.15 ^de^	28.11 ± 1.54 ^l^
*Centella asiatica*	16.31 ± 0.41 ^j^	14.74 ± 0.62 ^k^	37.88 ± 2.37 ^jk^
*Citrus aurantium*	16.92 ± 1.14 ^j^	27.92 ± 0.99 ^f^	53.87 ± 3.74 ^f^
*Coffea arabica*	90.72 ± 0.34 ^c^	99.63 ± 0.41 ^b^	98.36 ± 0.96 ^b^
*Curcuma longa*	108.36 ± 2.35 ^b^	102.99 ± 0.23 ^a^	123.77 ± 0.72 ^a^
*Daucus carota*	24.18 ± 0.38 ^g^	17.43 ± 1.19 ^j^	35.84 ± 2.75 ^k^
*Ganoderma lucidum*	4.79 ± 0.21 ^l^	10.98 ± 1.06 ^l^	40.64 ± 1.65 ^ij^
*Garcinia mangostana*	30.80 ± 1.45 ^f^	69.18 ± 3.91 ^c^	59.45 ± 5.39 ^e^
*Gynostemma pentaphyllum*	3.35 ± 0.24 ^l^	20.59 ± 1.26 ^i^	42.50 ± 2.16 ^hi^
*Kaempferia parviflora*	5.52 ± 0.28 ^l^	6.40 ± 0.49 ^m^	47.85 ± 1.93 ^g^
*Matricaria chamomilla*	18.31 ± 1.09 ^ij^	24.71 ± 2.13 ^gh^	45.77 ± 1.66 ^gh^
*Moringa oleifera*	12.31 ± 0.38 ^k^	18.73 ± 0.91 ^ij^	38.36 ± 1.94 ^jk^
*Piper nigrum*	20.61 ± 1.09 ^hi^	27.10 ± 1.07 ^fg^	75.31 ± 0.74 ^d^
*Tagetes erecta*	113.19 ± 6.95 ^a^	42.23 ± 3.50 ^e^	19.74 ± 0.38 ^m^
*Terminalia chebula*	78.52 ± 2.15 ^d^	99.55 ± 0.08 ^b^	95.20 ± 1.05 ^b^
*Zingiber officinale*	22.49 ± 0.96 ^gh^	23.15 ± 1.73 ^h^	24.34 ± 1.69 ^l^

^a–n^: Values in the same row with different superscripts are significantly different (*p* < 0.05).

**Table 5 nutrients-17-00898-t005:** The antioxidant capacities of promising standardized extracts from the list of Thai FDA-approved dietary supplements derived from medicinal plants.

Medicinal Plants	IC_50_; mg/mL			
	Metal Chelating ^1^	DPPH ^2^	ABTS ^3^	NBT ^4^
*Bacopa monnieri*	0.21 ± 0.01	0.38 ± 0.10 ^a^	2.82 ± 0.29 ^b^	0.21 ± 0.10 ^bc^
*Camellia sinensis*	1.23 ± 0.31	0.55 ± 0.03 ^a^	2.97 ± 0.24 ^b^	0.06 ± 0.02 ^a^
*Coffea arabica*	1.19 ± 0.19	0.17 ± 0.00 ^a^	0.12 ± 0.01 ^a^	0.29 ± 0.13 ^cd^
*Curcuma longa*	0.06 ± 0.00	0.42 ± 0.02 ^a^	0.18 ± 0.01 ^a^	038 ± 0.07 ^d^
*Tagetes erecta*	0.91 ± 0.07	12.40 ± 3.03 ^b^	8.73 ± 1.00 ^d^	0.06 ± 0.01 ^a^
*Terminalia chebula*	0.41 ± 0.01	0.98 ± 0.03 ^a^	6.43 ± 0.13 ^c^	0.13 ± 0.06 ^ab^

Note ^1^: Preventive properties metal-chelating activity (MCA) of extracts are expressed as the amounts that inhibited the ferrous ion-ferrozine complex by 50% (IC_50_; mg/mL), ^2,3^: Free radical scavenging activities of extracts are expressed as the concentration caused 50% inhibition of DPPH and ABTS+ radicals (IC_50_; mg/mL), and ^4^: the superoxide anion radical scavenging activity tested by the nitroblue tetrazolium (NBT) assay are expressed as the concentration that caused 50% inhibition of superoxide anion radicals (IC_50_; mg/mL). ^a–d^: Values in the same row with different superscripts are significantly different (*p* < 0.05).

## Data Availability

All data generated and analyzed are included in this research article. Additional data is available upon request to the corresponding author.

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
