# Peer review of "Exploring Antioxidant Properties of Standardized Extracts from Medicinal Plants Approved by the Thai FDA for Dietary Supplementation"

_nutrients, 2025, doi:10.3390/nu17050898_

Round 1

Reviewer 1 Report

Comments and Suggestions for Authors

This manuscript explores the growing interest in plant-derived antioxidants as functional food ingredients, particularly for their potential in combating neurodegenerative diseases. The researchers systematically reviewed 211 standardized extracts, narrowing down to 21 based on antioxidant and acetylcholinesterase inhibitory activities. Six extracts, including Bacopa monnieri, Camellia sinensis, and Curcuma longa, were further evaluated for their phenolic and flavonoid content and antioxidant properties using various assays. The results showed high antioxidant activity, with Bacopa monnieri, Coffea arabica, and Curcuma longa standing out as promising neuroprotective agents. The study highlights the potential of these extracts in the functional food industry but also emphasizes the need for further research on bioavailability, safety, and consumer acceptance.

1.       Please rewrite the abstract as it is too long. Please briefly include what the manuscript is about. Remove the conclusion from the abstract.

2.       What specific criteria were used to select the 21 medicinal plants from the initial 211? Were there any exclusion criteria that could have impacted the results?

3.       Please make italic in vivo and in vitro in the text. Revise all the manuscript. 

4.       I noticed that the introduction part does not clearly articulate why this article is useful and necessary. To strengthen your manuscript, I recommend providing a more compelling argument that reveals the significance of your research, its contribution to the field, and the gap it aims to fill.

5.       Please where you are using the text Latin names of plants make them italic.

6.       Please put the charges of the radicals as superscript. For example ABTS•+ -> ABTS•+. Please revise all manuscript.

7.       In the conclusion, focus on summarizing the key results rather than explaining why the research is important. The current text of the conclusion is more suited for the introduction section.

8.       How do the antioxidant properties of these extracts compare with existing commercial antioxidants? Are there any benchmarks used for comparison?

The manuscript I reviewed is well-organized, with clearly explained experiments that were done correctly, showing the authors' knowledge in the field.

Author Response

Reviewer #1:

Reviewer #1-1 Please rewrite the abstract as it is too long. Please briefly include what the manuscript is about. Remove the conclusion from the abstract. 

Response: The abstract has undergone revisions as recommended, resulting in a total of 221 words.

Reviewer #1-2 What specific criteria were used to select the 21 medicinal plants from the initial 211? Were there any exclusion criteria that could have impacted the results?

Response:

The inclusion criteria were: (I) plants cultivated domestically were included to ensure the local availability of herbal products; (II) their efficacy in scavenging free radicals was considered, taking into account their role in the onset of neurodegenerative diseases, evaluated using widely accepted methods such as DPPH and ABTS assays; (III) the inhibitory capacity against acetylcholinesterase (AChE) was examined, which is a recognized screening method for plants with neuroprotective potential; and (IV) commercially available standardized extracts were included to assess the practicality of the identified medicinal plants for use in the functional food industry.  The mentioned information was given in the Materials and Methods section.

       However, as the reviewer suggested, the exclusion criteria may have impacted the results, especially since the plants were selected based on their availability as standardized extracts in the market. Some plants, such as Alpinia galanga, Beta vulgaris, Cynara scolymus, Fragaria ananassa, Garcinia atroviridis, Illicium verum, and Origanum vulgare, which have been reported to possess a notable AChE inhibitory effect, were excluded from this study due to the lack of commercially available extracts. This addition has been made in the revised manuscript, as suggested.      

Reviewer #1-3 Please make italic in vivo and in vitro in the text. Revise all the manuscript. 

Response: In vivo and in vitro were italicized throughout the documents.

Reviewer #1-4, I noticed that the introduction part does not clearly articulate why this article is useful and necessary. To strengthen your manuscript, I recommend providing a more compelling argument that reveals the significance of your research, its contribution to the field, and the gap it aims to fill. 

       Response: The introduction has been revised for improved clarity, as recommended.

 “Thailand, known for its rich biodiversity and deep historical background in traditional medicine, has a substantial collection of medicinal plants recognized for their health-promoting properties. Additionally, the significant market in Thailand for food supplements and functional foods, coupled with its increasing aging population [17, 18], requires more research to focus on locally based plants approved by the Thai FDA for dietary supplements. The Thai FDA has granted approval to several plant-based extracts for dietary supplementation [16]. However, their potential as functional ingredients remains underexplored. Though previous studies have shown the antioxidant and neuroprotective properties of specific medicinal plants, a comprehensive and systematic evaluation of Thai FDA-approved standardized extracts for their potential use in functional food development is lacking. Furthermore, challenges such as standardization, quality control, and consumer acceptance hinder the wider application of these extracts in commercial settings products.

This study aims to address the existing gap by systematically evaluating the antioxidant properties of selected medicinal plants that have received approval from the Thai FDA for dietary supplements. By assessing the phenolic and flavonoid content, metal chelation activity, and free radical scavenging potential of these plants, we can identify promising candidates for incorporation into health-promoting functional foods. Furthermore, considering the significant role of oxidative stress in neurodegenerative diseases, this research establishes a foundational framework for using these extracts as neuroprotective agents. By bridging the gap between traditional knowledge and scientific validation, our findings contribute to the development of evidence-based functional food products designed to support brain health and overall well-being The findings of this study not only advance the scientific understanding of these extracts but also have practical implications for public health and the nutraceutical market.

Reviewer #1-5, Please where you are using the text Latin names of plants make them italic. 

       Response: The scientific name of the plant has been edited in italics.

Reviewer #1-6, Please put the charges of the radicals as superscript. For example ABTS•+ -> ABTS•+. Please revise all manuscript.

       Response: The charge of the radical has been adjusted to superscripts, and the chemical formula has been revised accordingly subscripts.

Reviewer #1-7, In the conclusion, focus on summarizing the key results rather than explaining why the research is important. The current text of the conclusion is more suited for the introduction section. 

       Response: The conclusion has been revised for improved clarity, as recommended.

“This study conducted a systematic evaluation of the antioxidant properties of Thai FDA-approved medicinal plant extracts, identifying Bacopa monnieri, Camellia sinensis, Coffea arabica, Curcuma longa, Tagetes erecta, and Terminalia chebula as the most promising candidates. These extracts displayed high levels of phenolic and flavonoid compounds, strong metal-chelating ability, and significant free radical scavenging activity, indicating their potential in functional food applications. The results provide valuable insights into the antioxidant capabilities of these plant-derived substances, laying a scientific groundwork for their use in nutraceutical and functional food products. Future research should prioritize bioavailability, long-term safety, and formulation strategies to enhance their practical application in consumer health items.”

Reviewer #1-8, How do the antioxidant properties of these extracts compare with existing commercial antioxidants? Are there any benchmarks used for comparison? 

       Response: The antioxidant capacities of these extracts were compared with those of butylated hydroxytoluene, ascorbic acid, α-tocopherol, and quercetin. The information was added to the Discussion section.

“Although their antioxidant capacities are lower than those of commercially available antioxidants such as butylated hydroxytoluene, ascorbic acid, α-tocopherol, and quercetin, these plants have been reported to exhibit AChE inhibitory effects, which are considered fundamental for the onset of neurodegenerative diseases.”

Reviewer 2 Report

Comments and Suggestions for Authors

The Authors presented the selected properties of the selected plants/plant extract. The purpose of the study is not highlighted and does not correspond with the title as well as an introduction. Generally, the manuscript content is difficult to understand, due to not a clearly presented methodology.

Please find below detailed comments which might be helpful for further improvements:

Abstract

- line 18 - Is "combat" is needed?

- l. 19-20 Highlight which plants (name it)

- l. 27 - which one (extracts)

please change general information into detailed

cross-out subsections in the abstract

conclusions are not needed here - give just 1-2 sentences as conclusions/perspectives

Introduction

- it is quite difficult to find the connection with subsection one about the economic 

- same comment for subsection 2

General remark:

- there is a lack of information as to why the Authors have chosen selected extracts. In my opinion, the whole introduction should be rewritten and highlight the most important information about the properties, and possible applications of selected extracts.

Materials and methods

- please add the manufacturer of the standardized extract

- subsection 2 is difficult to understand. Maybe it will be easier if you add the methodology of selection

- other parts are quite well organized

Results and discussion

- please use colors - the manuscript content will be more visible

- it is difficult to catch the idea and methodology of the plant/main compound selection even if some scheme occurred

- please summarize what kind of plants were used and the methods of the extract extraction - did you use commercially available compounds or did you take from the plants on your own

Author Response

Reviewer #2:

       The Authors presented the selected properties of the selected plants/plant extract. The purpose of the study is not highlighted and does not correspond with the title as well as an introduction. Generally, the manuscript content is difficult to understand, due to not a clearly presented methodology.

Please find below detailed comments which might be helpful for further improvements:

Reviewer #2-1-Abstract:

- line 18 - Is "combat" is needed?

- l. 19-20 Highlight which plants (name it)

- l. 27 - which one (extracts)

- please change general information into detailed

- cross-out subsections in the abstract

- conclusions are not needed here - give just 1-2 sentences as conclusions/perspectives

Response: The abstract has been revised for clarity, and the subsections have been crossed out as suggested.

“There is a growing interest in plant-derived antioxidants as functional food ingredients, given their potential to address oxidative stress-related diseases, notably neurodegenerative disorders. This study aims to investigate the antioxidant properties of medicinal plants that have been approved by the Thai FDA for dietary supplementation, with the goal of further utilizing them as dietary aids to prevent neurodegenerative conditions. A systematic review-based methodology was employed to select 21 out of 211 standardized extracts based on their documented antioxidant activity and acetylcholinesterase (AChE) inhibitory capacity. The 21 chosen standardized extracts were subjected to evaluation for their phenolic and flavonoid content, as well as their antioxidant activities utilizing metal chelating activity, DPPH, ABTS free radical scavenging, ferric reducing antioxidant power (FRAP), and superoxide anion scavenging techniques. Results: Among 21 plant extracts, six extracts—Bacopa monnieri, Camellia sinensis, Coffea arabica, Curcuma longa, Tagetes erecta, and Terminalia chebula—were found to exhibit promising antioxidant capacities. The selected extracts demonstrated elevated levels of phenolics (up to 1,378.19 mg gallic acid equivalents per gram) and flavonoids, in addition to potent antioxidant activities. Notably, Coffea arabica and Curcuma longa exhibited particularly enhanced antioxidant and free radical scavenging capabilities, underscoring their potential as neuroprotective agents. Due to their high levels of phenolic and flavonoid compounds, along with strong metal-chelating abilities and significant free radical scavenging activities, these standardized extracts show potential for functional food applications that may help delay the onset of neurodegenerative diseases.”

Introduction:

  • It is quite difficult to find the connection with subsection one about the economic. 
  • -There is a lack of information as to why the Authors have chosen selected extracts. In my opinion, the whole introduction should be rewritten and highlight the most important information about the properties, and possible applications of selected extracts.

Response: The introduction section has been rewritten to enhance clarity and emphasize the significant gaps in knowledge for this study, as suggested.

“Thailand, known for its rich biodiversity and deep historical background in traditional medicine, has a substantial collection of medicinal plants recognized for their health-promoting properties. Additionally, the significant market in Thailand for food supplements and functional foods, coupled with its increasing aging population [17, 18], requires more research to focus on locally based plants approved by the Thai FDA for dietary supplements. The Thai FDA has granted approval to several plant-based extracts for dietary supplementation [16]. However, their potential as functional ingredients remains underexplored. Though previous studies have shown the antioxidant and neuroprotective properties of specific medicinal plants, a comprehensive and systematic evaluation of Thai FDA-approved standardized extracts for their potential use in functional food development is lacking. Furthermore, challenges such as standardization, quality control, and consumer acceptance hinder the wider application of these extracts in commercial settings products.

This study aims to address the existing gap by systematically evaluating the antioxidant properties of selected medicinal plants that have received approval from the Thai FDA for dietary supplements. By assessing the phenolic and flavonoid content, metal chelation activity, and free radical scavenging potential of these plants, we can identify promising candidates for incorporation into health-promoting functional foods. Furthermore, considering the significant role of oxidative stress in neurodegenerative diseases, this research establishes a foundational framework for using these extracts as neuroprotective agents. By bridging the gap between traditional knowledge and scientific validation, our findings contribute to the development of evidence-based functional food products designed to support brain health and overall well-being The findings of this study not only advance the scientific understanding of these extracts but also have practical implications for public health and the nutraceutical market.”

Materials and methods:

  • please add the manufacturer of the standardized extract
  • subsection 2 is difficult to understand. Maybe it will be easier if you add the methodology of selection
  • other parts are quite well organized

Response: All standard extract manufacturers have been added. The subsection 2 was revised as follows.

“The medicinal plants examined in this study (n=211) were thoroughly documented in the approval list by the Thai FDA for their use as dietary supplements [16]. To conduct a comprehensive exploration of locally sourced functional ingredients with neuroprotective properties, specific inclusion criteria were followed in selecting candidate medicinal plants. The first inclusion criterion was that only domestically cultivated plants were considered to ensure the local availability of herbal products. Secondly, the plants were selected based on their reported effectiveness in scavenging DPPH and ABTS radicals, which are well-established methodologies. The rationale for selection included the widely recognized fact that oxidative damage from excess free radicals significantly contributes to neurodegenerative diseases. Third, available information on their acetylcholinesterase (AChE) inhibitory capacity, a recognized screening approach for plants with neuroprotective potential, was included in the selection criteria. Finally, plants with commercially available standardized extracts were considered to verify their feasibility for use in the functional food industry. Detailed information on the 211 medicinal plants, including their scientific names, utilized plant parts, antioxidant properties, and AChE inhibitory effects, is provided in Supplementary Tables 2, 3, and 4, respectively.”

Results and discussion:

  • please use colors - the manuscript content will be more visible
  • it is difficult to catch the idea and methodology of the plant/main compound selection even if some scheme occurred
  • please summarize what kind of plants were used and the methods of the extract extraction - did you use commercially available compounds, or did you take from the plants on your own

Response: Figures 2(b), 2(d), 2(e), and 2(f) have been updated to color. The materials and methods section, as well as the results section, have been revised as highlighted to enhance clarity.

Round 2

Reviewer 1 Report

Comments and Suggestions for Authors

Authors have corrected the manuscript regarding my comments and recommendations. 

Author Response

Dear Reviewer, 

we are grateful for the opportunity to revise our manuscript and contribute valuable information on Nutrients to the scientific community. We sincerely appreciate your consideration and support.

Best Regards,

Reviewer 2 Report

Comments and Suggestions for Authors

The manuscript needs major revision. Please find detailed comments below:

Abstract

  • In this form sounds much better. However still it is not clear do you want to perform a review paper or if you investigated selected extracts (which one how many - you can add only a number). Highlight which one/ones are most promising and show the best desirable properties

Introduction

  • in the last section please add information on how many extracts you investigate

Materials and methods

  • please show all 21 standardized extracts in the main body text - if you place them in the table i.e. name/compound/supplier, etc. it will be clearer to see what was investigated
  • other parts are quite clear and can be as is

Results and discussion

  • figure 1 is more suitable for materials and methods in my opinion. Please respond as to why you want to present it here
  • figure 2 Unfortunately it is not clear to me what the Authors wanted to present here
  • same comment for figure 3 - what do you mean by heat map? Furthermore, when you give a short name it should be presented before (see remark about adding additional table in materials and methods)
  • clarify what you mean by standardized extracts
  • Table 4 - why there are not all of the investigated samples - please clarify it
  • personally, I will make results and discussion sections common - it will be more clear to understand the methodology and presented results, however, it is up to the Authors

Author Response

Abstract

  • In this form sounds much better. However still it is not clear do you want to perform a review paper or if you investigated selected extracts (which one how many - you can add only a number). Highlight which one/ones are most promising and show the best desirable properties
  • Response: The abstract has undergone revisions as recommended. The promising plants were highlighted with yellow as follows.

“There is a growing interest in plant-derived antioxidants as functional food ingredients, given their potential to address oxidative stress-related diseases, notably neurodegenerative disorders. This study aims to investigate the antioxidant properties of medicinal plants that have been approved by the Thai FDA for dietary supplementation, with the goal of further utilizing them as food-functional ingredients to prevent neurodegenerative conditions. A systematic review-based methodology was employed on listed 211 medicinal plants and 21 medicinal plants were chosen based on their documented antioxidant activity and acetylcholinesterase (AChE) inhibitory capacity. The 21 commercially available standardized extracts were subjected to evaluation for their phenolic and flavonoid content, as well as their antioxidant activities utilizing metal chelating activity, DPPH, ABTS free radical scavenging, ferric reducing antioxidant power (FRAP), and superoxide anion scavenging techniques. Among the 21, six extracts—Bacopa monnieri, Camellia sinensis, Coffea arabica, Curcuma longa, Tagetes erecta, and Terminalia chebula—emerged as the most promising. These extracts exhibited elevated levels of phenolic (up to 1,378.19 mg gallic acid equivalents per gram) and flavonoid compounds, with Coffea arabica and Curcuma longa showing the strongest antioxidant and free radical scavenging activities, indicating their potential for use in functional foods aimed at delaying neurodegenerative diseases. Due to their high levels of phenolic and flavonoid compounds, along with strong metal-chelating abilities and significant free radical scavenging activities, these standardized extracts show potential for functional food applications that may help delay the onset of neurodegenerative diseases.”

Introduction

  • in the last section please add information on how many extracts you investigate.
  • Response: The information was added and highlighted with yellow as follows.

“This study aims to address the existing gap by systematically evaluating the antioxidant properties of 21 selected medicinal plants that have received approval from the Thai FDA for dietary supplements.”

Materials and methods

  • Please show all 21 standardized extracts in the main body text - if you place them in the table i.e. name/compound/supplier, etc. it will be clearer to see what was investigated
  • Response: Table 1 was added to provide in mentioned information.
  • other parts are quite clear and can be as is

Results and discussion

  • figure 1 is more suitable for materials and methods in my opinion. Please respond as to why you want to present it here.
  • Response: Figure 1 was moved to the materials and methods section as suggested. 
  • figure 2 Unfortunately it is not clear to me what the Authors wanted to present here
  • Response: Figure 2 was added to describe the characteristics of 211 medicinal plant extracts listed in the Thai FDA approval list and provide detial how 21 medicinal plants were chosen for further experiments.

  • same comment for figure 3 - what do you mean by heat map? Furthermore, when you give a short name it should be presented before (see remark about adding additional table in materials and methods)
  • Response: The heat map was added to summarize and clearly illustrate why six medicinal plants were highlighted as promising extracts for potential use as functional food ingredients. The corresponding short names mentioned in the heat map have been included in Table 2.

  • clarify what you mean by standardized extracts
  • Response: The standardized extracts refer to extracts that are commercially available with known active constituents, in accordance with the requirements outlined by the Thai FDA.
  • Table 4 - why there are not all of the investigated samples - please clarify it
  • Response: Based on the information presented in the heat map, only six medicinal plants were selected as promising extracts and further tested for potential use as functional food ingredients. Therefore, only these six extracts are included in Table 4.

Round 3

Reviewer 2 Report

Comments and Suggestions for Authors

Dear Authors
Thank you for your updates and responses.
In its present form, the manuscript looks much, much better and is almost ready.
Based on the presented results and your comments it looks like you made 2 selections first from over 200 to 21 compositions from which you recommended six. Please have a look at the flow chart as well as the abstract. In my opinion, you should add information that the final six "the best" should be presented (add this information)
Still, it is difficult for me to understand the heat chart, especially what you mean by value 0-100.
My recommendation is a minor revision.

Author Response

Thank you for giving us the opportunity to submit a revised draft of our manuscript, "Exploring antioxidant properties of standardized extracts from medicinal plants approved by the Thai FDA for dietary supplementation.” The revised version includes only two minor edits to Figures 1 and 3. Furthermore, the figure legend for Figure 3 has been revised to improve clarity.

Once again, we are grateful for the opportunity to revise our manuscript and contribute valuable information on Nutrients to the scientific community. We sincerely appreciate your consideration and support.

Thank you for considering our submission.

Sincerely,
